# Tunneled peripherally inserted central catheter versus non-tunneled and its effects in clinical outcomes: A multicenter randomized clinical trial protocol

Rodrigo do Nascimento Ceratti[1], Knut Taxbro[2,3], Vineet Chopra[4], Leandro Augusto Hansel[1], Carolina Geske Salini[1], Ivana Duarte Brum[1], Marina Junges[1], Arlene Gonçalves dos Santos[1], Eneida Rejane Rabelo-Silva [5,6,*]

1 Hospital de Clínicas de Porto Alegre, Vascular Access Program, Rio Grande do Sul, Brazil, 2 Linköping University, Linköping, Sweden, 3 Department of Biomedical and Clinical Sciences, Linköping University, Linköping, Sweden, 4 Department of Medicine, University of Colorado Anschutz Medical Campus, Aurora, Colorado, United States of America, 5 Universidade Federal do Rio Grande do Sul, Porto Alegre, Brazil, 6 Department of Anaesthesia and Intensive Care Medicine, Ryhov County Hospital, Jönköping, Sweden

☉ These authors contributed equally to this work.
* eneidarabelo@gmail.com

## Abstract

### Background

The use of peripherally inserted central catheters (PICC) has increased due to its benefits, such as greater durability, safety, comfort, and cost-effectiveness. Technological advancements, such as catheter tip navigation systems and the use of ultrasound, have improved its quality. However, complications still occur, including infections and thrombosis, especially in oncology and intensive care patients. Studies indicate that advanced practices, technology, and specialized teams reduce these risks. New techniques, such as tunneled insertion, show potential for reducing complications, but further research with larger samples is needed to validate these findings.

### Objective

To compare the tunneling technique of PICC to non-tunneling insertion technique regarding the incidence of isolated or combined outcomes of catheter-related bloodstream infection, thrombosis, occlusion, and accidental dislodgement in the adult population within a 30-days period.

### Materials and methods

In this randomized, parallel, multicenter clinical trial, 840 patients from three reference hospitals will be assigned to two parallel groups (conventional PICC and tunneled PICC groups) through computer-generated stratified randomization. The

**Data availability statement:** Deidentified research data will be made publicly available when the study is completed and published.

**Funding:** This study was partially funded by the Coordination for the Improvement of Higher Education Personnel, Brazil (Sandwich Doctorate, number 88887.936644/2024-00). This study was also supported by the National Council for Scientific and Technological Development (number 03473/2025-8) and the Research Incentive and Fund of the Hospital de Clínicas de Porto Alegre (https://ror.org/010we4y38) (number 2023-0232). The funders were not involved in the study design; data collection, analysis, or interpretation; or manuscript writing.

**Competing interests:** The authors have declared that no competing interests exist.

conventional group will undergo PICC insertion according to routine practice. In the tunneled PICC group, an additional subcutaneous tunneling procedure will be performed. Patients will be followed until PICC removal for any reason or 30 days after insertion, whichever occurs first. The primary outcome is to assess whether subcutaneous tunneling reduces the rate of isolated or combined adverse events (infection, thrombosis, obstruction, and dislodgement) compared to the conventional method.

## Discussion

Subcutaneous tunneling is a widely used method to reduce complications associated with catheters. However, its application in PICC has not yet been extensively explored, especially in Brazil. A randomized clinical trial is necessary to objectively assess the effects of subcutaneous tunneling in PICC insertion. This protocol aims to provide evidence on the effectiveness of this technique in reducing complications.

## Trial registration

Clinical Trials platform NCT06365528

---

## Introduction

The use of peripherally inserted central catheters (PICCs) has grown in recent years due to their advantages over other catheters. Research highlights benefits such as longer dwell times, durability, safety, comfort, versatility [1,2], and cost-effectiveness [3]. Industry advancements in materials (polyurethane, polyethylene) and technology have further driven adoption. Innovations include tip navigation and location systems [4], ultrasound, micro-introduction, and the Modified Seldinger Technique [5,6].

However, despite all these technical advancements and the deepening of knowledge related to the use of PICC, this device is not free from risks. Major complications, such as thrombosis and infection, and minor complications, such as phlebitis, accidental traction, and catheter displacement, can occur at varying rates depending on the study population. In adults, studies indicate catheter-related infection rates ranging from 0.07 to 2.46 per 1000 catheter-days, which are more common in oncology and intensive care patients [7,8].

Regarding thrombosis, patients with hematologic cancer are more predisposed to this condition, with incidence rates varying from 2% to 75% [9–12]. However, it is noteworthy that these rates can be reduced with the use of higher-quality devices, advanced technology, vascular access specialist nurses, and the adoption of best practices in both insertion and maintenance [13]. A multicenter study conducted in 16 centers in Brazil on patients with PICCs supports these findings, showing low complication rates, such as catheter-related bloodstream infection, venous thrombosis, and reversible occlusion [14].

While technology, vascular access teams, best practices, and infection control bundles have reduced PICC-related complications, new insertion techniques can further minimize risks. Studies support the tunneling technique over conventional

insertion [15,16]. A randomized clinical trial (RCT) found tunneled PICCs had longer dwell times and lower thrombosis and infection rates [16], with another RCT confirming reduced thrombosis and accidental tractions [17].

Based on this, we understand that there is a lack of studies with robust methodologies, especially those with larger sample sizes, to corroborate the current results. Similarly, few studies demonstrate the combined impact of tunneled PICC insertion on reducing negative outcomes. Therefore, we test the hypothesis that tunneled PICCs have lower rates of the combined or isolated outcomes like catheter-related bloodstream infections, thrombosis, occlusion, and accidental traction compared to PICC inserted using the conventional technique in the adult population of three large hospitals involved in the RCT.

## Methods

### Study design

This is a randomized, parallel, multicenter clinical trial, blinded to the allocation groups during the analysis of the primary outcome, conducted in three hospitals in Brazil. This study has been registered on the Clinical Trials platform under NCT06365528.

### Participants and eligibility criteria

The study will include adult patients (≥ 18 years) admitted to medical-surgical and intensive care units with an indication for PICC, regardless of the number of lumens or catheter gauge. Patients will be excluded from the study if they do not have suitable venous conditions for safe PICC insertion, which means occupancy rate >45% in the green zone (according to the Zone Insertion Method) [18]; patients with chronic kidney disease, whether on dialysis or not; patients in a critical or unstable condition; or patients with cognitive deficits that impair their understanding of the study and who do not have a responsible person to assist at this stage.

### Settings and location

This multicenter RCT will be conducted in three hospitals in Porto Alegre, RS, Brazil (Hospital de Clínicas de Porto Alegre, Santa Casa de Misericórdia de Porto Alegre and Hospital Moinhos de Vento), from May 2024 to December 2025. All centers are large, tertiary hospitals that offer the best care for various health conditions. Additionally, they have dedicated teams of nurses specializing in vascular access and infusion therapy.

### Groups and Interventions Intervention description

The patients recruited for the study will be randomly divided into two groups: Intervention Group (IG) – adult patients who will have the tunneled PICC insertion; and Control Group (CG) – adult patients who will have the PICC insertion performed using the conventional technique (non-tunneled).

When an adult patient requires a peripherally inserted central catheter (PICC), the medical or nursing team will notify researchers via an electronic system or medical prescription, providing patient details. The research team will assess eligibility, invite qualifying patients, and obtain informed consent. Participants will be randomly assigned to: a) PICC insertion using the tunneling technique (Intervention Group), or b) PICC insertion using the conventional technique (Control Group). Non-participants' data will be stored in a database of potentially eligible patients. Research team nurses with tunneling expertise will perform the Intervention Group procedures, while hospital nurses unaffiliated with the research team will handle the Control Group procedures.

The catheters inserted in the Control Group will follow the routine procedure of each institution, according to their respective Standard Operating Procedures (SOP), with ultrasound-guided puncture and modified Seldinger technique. The procedures for the Intervention Group, in addition to the conventional PICC insertion technique, will include the additional steps of catheter tunneling, as described below (S2 Fig):

## Outcomes

**Catheter-related primary bloodstream infection:** bloodstream infection in a patient using a peripherally inserted central catheter (PICC) for more than two days (with the first day being the day of catheter insertion) and diagnosed with infection either while the catheter is still in use or within one day after its removal [19].
**Catheter-related thrombosis:** formation of thrombi inside the vein leading to obstruction of blood flow. Diagnosis should be made using appropriate imaging exams (venous ultrasound and/or venous Doppler), associated or not with the following signs and symptoms: pain, swelling, redness, dilated superficial veins, impaired limb movement, low-grade fever [20].
**Irreversible catheter occlusion:** Catheter occlusion is characterized by the blockage of the lumen due to the formation of a blood clot or precipitate of medications, preventing the infusion of intravenous solution [20].
**Accidental catheter dislodgement:** Characterized by the accidental partial or total exteriorization of the catheter by a healthcare team member or the patient themselves, rendering the catheter unusable. Inviability of the catheter is defined by the improper positioning of the distal end of the catheter outside the cavoatrial junction, where positioning of the catheter is mandatory [20].

## Participant timeline

See S3 Table.

## Sample size

The sample size calculation considered a composite outcome including primary bloodstream infection related to the catheter, thrombosis, catheter obstruction, and accidental catheter traction. Sample size was calculated to detect differences in proportions between the control and intervention groups using the online tool PSS Health version [21], with a statistical power of 80%, a significance level of 5%, and a relative risk of 0.5 [22], indicating a 50% reduction in composite outcomes. The calculated sample size, with an additional 10% for potential losses, totals 840 subjects, divided equally (420 each) between the two groups across the three institutions.

## Recruitment

When an adult patient needs a peripherally inserted central catheter (PICC), the medical or nursing team will notify researchers via electronic systems or medical prescriptions with patient details. The research team will assess eligibility, invite qualifying patients, and obtain informed consent. Participants will be randomly assigned to study groups. Recruitment started on May 6, 2024 and will be completed on December 31, 2025. The data collection will be finished in 30 days after the last inclusion.

## Randomization Sequence generation

Participants, after giving informed consent, will be randomly assigned to either the Intervention Group or the Control Group. Randomization will be generated using a dedicated tool within the REDCap software, in blocks of varying sizes. Each participating center will manage the randomization of their participants. The strategy of varying block sizes will be employed to ensure allocation concealment.

## Implementation

The allocation sequence of participants in the study will be digitally and automatically conducted using the REDCap software. One researcher from each participating center, blinded to the data collection phase of the research, will be responsible for generating the sequence. Participant enrollment in the study will be carried out by the researchers responsible for implementing the study at each participating center.

 

## Blinding

After group assignment, the researchers at each participating center responsible for data collection, whether in person or virtually (via electronic medical records), will be blinded to the allocation of each participant. Study patients will be instructed by the research team that conducted the allocation not to disclose which intervention was received.

The researchers responsible for outcome analysis, as well as the statistical team involved in the analysis, will be blinded since they will not be involved in the recruitment, allocation, or data collection stages.

## Statistical methods

The data will be entered and analyzed using the Statistical Package for the Social Sciences (SPSS) version 24.0. Initially, a Shapiro-Wilk test will be conducted to assess the normality of continuous variable distributions. Following this stage, a descriptive analysis will be performed, in which results will be reported as mean ± standard deviation (SD), median (P25 - P75), or as absolute and relative frequencies, depending on the characteristics and distribution of the variables.

The quantitative variables between groups will be compared using either Student's t-test or the Mann-Whitney U test, depending on the data distribution. Pearson's Chi- square test will be employed to assess associations between the clinical characteristics of patients and devices with events during follow-up. Groups will be compared for device event-free survival using Cox proportional hazards analysis and the log-rank test.

Other associations will be analyzed based on data distribution and appropriate statistical tests. A two-tailed $P < 0.05$ will be considered statistically significant.

## Research ethics approval

This study has been approved by the Committees of Ethics in Researches (CER)/Institutional Review Board (IRB) of the respective participating institutions on August 22, 2023 by the number 71299723.0.1001.5327. Before random allocation, all patients or their legal representatives need to sign the Informed Consent Form (ICF) to ensure their acceptance to participate in the study.

## Discussion

Subcutaneous tunneling is a well-established technique that has been widely employed in various medical procedures to reduce the risk of complications associated with the placement of catheters. This method involves creating a tunnel beneath the skin through which the catheter is passed before entering the vein, thereby increasing the distance between the catheter's entry point into the vein and its exit site on the skin. This added distance can serve as a protective barrier against microbial contamination, potentially reducing the incidence of catheter-related infections and improving overall catheter stability.

Despite its proven benefits in other types of vascular access, the application of subcutaneous tunneling in PICCs remains underexplored, particularly within the Brazilian healthcare context. PICCs are commonly used for medium- to long-term intravenous therapy, especially in patients requiring extended antibiotic treatment, parenteral nutrition, or chemotherapy. Given their widespread use and the associated risk of complications such as infections, thrombosis, and catheter dislodgement, exploring new strategies to enhance their safety and effectiveness is of great clinical importance.

Currently, there is a lack of robust, high-quality evidence evaluating the impact of subcutaneous tunneling specifically in PICC placements. While unsystematic reports and smaller studies suggest potential benefits, the absence of large-scale randomized clinical trials limits the ability to draw definitive conclusions about the technique's effectiveness in this setting. This is especially true in Brazil, where there are significant regional differences in healthcare infrastructure, infection control practices, and patient populations.

To address this gap in the literature, a randomized clinical trial is necessary to objectively assess the outcomes of PICC insertion with and without the use of subcutaneous tunneling. Such a study would provide valuable data on whether this technique can indeed reduce complications such as infection, thrombosis, or catheter malfunction in Brazilian patients.

The protocol outlined here aims to establish a scientific basis for evaluating the use of subcutaneous tunneling in PICC insertion. By systematically comparing outcomes between tunneled and non-tunneled PICCs, this study intends to generate evidence that can inform clinical practice, contribute to the development of guidelines, and ultimately improve patient care by minimizing catheter-related complications.

## Supporting information

**S1 Fig. Research protocol flowchart.**
(TIF)

**S2 Fig. Tunneling technique.**
(TIF)

**S3 Table. Schedule of recruitment, interventions and assessment of SPIRIT (Standard Protocol Items: Recommendations for Interventional Trials) guidelines.**
(PDF)

## Author contributions

**Conceptualization:** Rodrigo do Nascimento Ceratti, Eneida Rejane Rabelo-Silva.

**Methodology:** Rodrigo do Nascimento Ceratti.

**Supervision:** Eneida Rejane Rabelo-Silva.

**Validation:** Knut Taxbro, Vineet Inder Chopra, Leandro Augusto Hansel, Carolina Geske Salini, Ivana Duarte Brum, Marina Junges, Arlene Gonçalves dos Santos.

**Writing – original draft:** Rodrigo do Nascimento Ceratti.

**Writing – review & editing:** Rodrigo do Nascimento Ceratti, Knut Taxbro, Vineet Inder Chopra, Leandro Augusto Hansel, Carolina Geske Salini, Ivana Duarte Brum, Marina Junges, Arlene Gonçalves dos Santos, Eneida Rejane Rabelo-Silva.

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
