## [Decision Letter · Decision Letter 0]

2 Nov 2025

Dear Dr. Rabelo-Silva,

Thank you for submitting your manuscript to PLOS ONE. After careful consideration, we feel that it has merit but does not fully meet PLOS ONE’s publication criteria as it currently stands. Therefore, we invite you to submit a revised version of the manuscript that addresses the points raised during the review process.

We look forward to receiving your revised manuscript.

Kind regards,

Madhuradhar Chegondi, MD

Academic Editor

PLOS ONE

Journal Requirements:

3. Please ensure that you refer to Figure 1 in your text as, if accepted, production will need this reference to link the reader to the figure.

4. We note you have included a table to which you do not refer in the text of your manuscript. Please ensure that you refer to Table 1 in your text; if accepted, production will need this reference to link the reader to the Table.

5. Please include your tables as part of your main manuscript and remove the individual files. Please note that supplementary tables (should remain/ be uploaded) as separate "supporting information" files

6. We note that the original protocol that you have uploaded as a Supporting Information file contains an institutional logo. As this logo is likely copyrighted, we ask that you please remove it from this file and upload an updated version upon resubmission.

Reviewers' comments:

Reviewer's Responses to Questions

**Comments to the Author**

1. Does the manuscript provide a valid rationale for the proposed study, with clearly identified and justified research questions?

Reviewer #1: Yes

Reviewer #2: Yes

2. Is the protocol technically sound and planned in a manner that will lead to a meaningful outcome and allow testing the stated hypotheses?

Reviewer #1: Yes

Reviewer #2: Partly

3. Is the methodology feasible and described in sufficient detail to allow the work to be replicable?

Reviewer #1: Yes

Reviewer #2: Yes

4. Have the authors described where all data underlying the findings will be made available when the study is complete?

Reviewer #1: Yes

Reviewer #2: No

5. Is the manuscript presented in an intelligible fashion and written in standard English?

Reviewer #1: Yes

Reviewer #2: Yes

You may also provide optional suggestions and comments to authors that they might find helpful in planning their study.

Reviewer #1: Interesting protocol

Some issues should be addressed

1) background in abstract shoudld be shortened

2)methods: it is not clear how randomization will be performed

3)sample calculation is not clear to me>authors should state if they start from some infections rates and go back to others. Moreover it should be performed on primary end point

4) do authors think that subgroup analysis should be performed

Reviewer #2: In this ambitious multicentric trial, the authors wish to investigate if tunneled PICC lines in adult patients provide benefit compared to standard PICC insertion techniques. In this trial, the authors wish to enroll 840 patients in three centers in Brazil and use stratified randomization. The primary outcome variable is the isolated or combined adverse events from PICC lines (infection, thrombosis, obstruction, or dislodgement).

Overall, the study design is pretty straightforward. In the supplemental protocol, the authors mentioned that the study team members inserting the tunnel for clients would be specifically trained. It is fairly ambitious with the amount of funding available to them. But if baseline insertions are part of standard patient management, then it should be feasible.

The authors’ sample size is based on an estimate of a 50% reduction in composite outcomes. This appears very, very optimistic. The authors have not provided any information on how they estimated a 50% reduction. This is important because an inadequately powered study (because of overestimation of potential benefit) with standard frequentist analysis is very likely to lead to a null effect.

The authors also mentioned that the research team would be blinded for tunneled vs. non-tunneled catheters. This is really not very feasible, as it is easy to identify tunneled vs. non-tunneled catheters. While blinding at the time of analysis can be ensured, data collection would have to be open-label, which is okay, as blinding is not realistic in this type of study.

The authors have also not mentioned any data collection on the success rate of inserting tunneled catheters and complications/costs associated with tunneled catheters compared to standard insertions. It seems like the authors wish to do an intention-to-treat analysis, which is okay. However, they should also do a per-protocol analysis as a sensitivity analysis.

**Do you want your identity to be public for this peer review?** For information about this choice, including consent withdrawal, please see our Privacy Policy

Reviewer #1: **Yes:** Fabrizio D'Ascenzo

Reviewer #2: **Yes:** Sandeep Tripathi

---

## [Author Response · Author response to Decision Letter 1]

15 Jan 2026

1. Please ensure that your manuscript meets PLOS ONE's style requirements, including those for file naming. OK

2. Please update your submission to use the PLOS LaTeX template. OK

3. Please ensure that you refer to Figure 1 in your text as, if accepted, production will need this reference to link the reader to the figure. Referred

4. We note you have included a table to which you do not refer in the text of your manuscript. Please ensure that you refer to Table 1 in your text; if accepted, production will need this reference to link the reader to the Table. Referred

5. Please include your tables as part of your main manuscript and remove the individual files. Please note that supplementary tables (should remain/ be uploaded) as separate "supporting information" files. OK

6. We note that the original protocol that you have uploaded as a Supporting Information file contains an institutional logo. As this logo is likely copyrighted, we ask that you please remove it from this file and upload an updated version upon resubmission. Removed

Reviewer #1:

background in abstract should be shortened

The background in abstract was shortened – Page 2

methods: it is not clear how randomization will be performed

The randomization was better described in methods – Page 7

sample calculation is not clear to me>authors should state if they start from some infection rates and go back to others. Moreover it should be performed on primary end point

Sample calculation was based on previous article, according to what is described in Sample size – Page 6

HONG, Jiana; MAO, Xiaodan. Complications of tunneled and non-tunneled peripherally inserted central catheter placement in chemotherapy-treated cancer patients: a meta-analysis. Frontiers in Surgery, v. 11, p. 1469847, 2024.

do authors think that subgroup analysis should be performed

We understand that a subgroup analysis could be performed, particularly with onco-hematologic patients. However, subgroup analysis will be conducted a posteriori.

Reviewer #2:

The authors’ sample size is based on an estimate of a 50% reduction in composite outcomes. This appears very, very optimistic. The authors have not provided any information on how they estimated a 50% reduction. This is important because an inadequately powered study (because of overestimation of potential benefit) with standard frequentist analysis is very likely to lead to a null effect.

Sample calculation was based on previous article, according to what is described in Sample size – Page 6;

HONG, Jiana; MAO, Xiaodan. Complications of tunneled and non-tunneled peripherally inserted central catheter placement in chemotherapy-treated cancer patients: a meta-analysis. Frontiers in Surgery, v. 11, p. 1469847, 2024.

The authors also mentioned that the research team would be blinded for tunneled vs. non-tunneled catheters. This is really not very feasible, as it is easy to identify tunneled vs. non-tunneled catheters. While blinding at the time of analysis can be ensured, data collection would have to be open-label, which is okay, as blinding is not realistic in this type of study.

Thank you for your observation. However, the professionals responsible for assessing the outcomes will not be directly involved with the patients. The presence or absence of each outcome will be determined based on medical records.

The authors have also not mentioned any data collection on the success rate of inserting tunneled catheters and complications/costs associated with tunneled catheters compared to standard insertions. It seems like the authors wish to do an intention-to-treat analysis, which is okay. However, they should also do a per-protocol analysis as a sensitivity analysis.

Thank you for your observation. All analyses will be performed following the intention-totreat principle. This is included in the analyses section.

---

## [Decision Letter · Decision Letter 1]

21 Jan 2026

TUNNELED PERIPHERALLY INSERTED CENTRAL CATHETER VERSUS NON-TUNNELED AND ITS EFFECTS IN CLINICAL OUTCOMES: A MULTICENTER RANDOMIZED CLINICAL TRIAL PROTOCOL

PONE-D-25-38258R1

Dear Professor Eneida Rejane Rabelo-Silva,

We’re pleased to inform you that your manuscript has been judged scientifically suitable for publication and will be formally accepted for publication once it meets all outstanding technical requirements.

Kind regards,

Madhuradhar Chegondi, MD

Academic Editor

PLOS One

Additional Editor Comments (optional):

Reviewers' comments:

Reviewer's Responses to Questions

**Comments to the Author**

1. Does the manuscript provide a valid rationale for the proposed study, with clearly identified and justified research questions?

Reviewer #1: Yes

2. Is the protocol technically sound and planned in a manner that will lead to a meaningful outcome and allow testing the stated hypotheses?

Reviewer #1: Yes

3. Is the methodology feasible and described in sufficient detail to allow the work to be replicable?

Reviewer #1: Yes

4. Have the authors described where all data underlying the findings will be made available when the study is complete?

Reviewer #1: Yes

5. Is the manuscript presented in an intelligible fashion and written in standard English?

Reviewer #1: Yes

You may also provide optional suggestions and comments to authors that they might find helpful in planning their study.

Reviewer #1: all comments have been addressed and authors should be complimented for this good revision again, although synthetic in answers

**Do you want your identity to be public for this peer review?** For information about this choice, including consent withdrawal, please see our Privacy Policy

Reviewer #1: **Yes:** Fabrizio D'Ascenzo

---

## [Editor Report · Acceptance letter]

PONE-D-25-38258R1

PLOS One

Dear Dr. Rabelo-Silva,

I'm pleased to inform you that your manuscript has been deemed suitable for publication in PLOS One. Congratulations! Your manuscript is now being handed over to our production team.

Kind regards,

on behalf of

Dr. Madhuradhar Chegondi

Academic Editor

PLOS One